# Insight into Antibiotic Synergy Combinations for Eliminating Colistin Heteroresistant *Klebsiella pneumoniae*

**DOI:** 10.3390/genes14071426

**Published:** 2023-07-10

**Authors:** Sahaya Glingston Rajakani, Basil Britto Xavier, Adwoa Sey, El Bounja Mariem, Christine Lammens, Herman Goossens, Youri Glupczynski, Surbhi Malhotra-Kumar

**Affiliations:** Laboratory of Medical Microbiology, Vaccine & Infectious Disease Institute, University of Antwerp, 2610 Antwerp, Belgium; sahayaglingston.rajakani@uantwerpen.be (S.G.R.); basilbritto@hotmail.com (B.B.X.); adwoasey31@gmail.com (A.S.); mariem.elbounja@student.uantwerpen.be (E.B.M.); christine.lammens@uantwerpen.be (C.L.); herman.goossens@uza.be (H.G.); gerald.glupczynski@uantwerpen.be (Y.G.)

**Keywords:** time–kill assay, population analysis profiling, PAP, polymyxin, combination therapies, multi-drug resistance, in vitro testing, heteroresistance

## Abstract

Colistin heteroresistance has been identified in several bacterial species, including *Escherichia coli* and *Klebsiella pneumoniae*, and may underlie antibiotic therapy failures since it most often goes undetected by conventional antimicrobial susceptibility tests. This study utilizes population analysis profiling (PAP) and time–kill assay for the detection of heteroresistance in *K. pneumoniae* and for evaluating the association between in vitro regrowth and heteroresistance. The mechanisms of colistin resistance and the ability of combination therapies to suppress resistance selection were also analysed. In total, 3 (18%) of the 16 colistin-susceptible strains (MIC ≤ 2 mg/L) were confirmed to be heteroresistant to colistin by PAP assay. In contrast to the colistin-susceptible control strains, all three heteroresistant strains showed regrowth when exposed to colistin after 24 h following a rapid bactericidal action. Colistin resistance in all the resistant subpopulations was due to the disruption of the *mgrB* gene by various insertion elements such as IS*Kpn14* of the IS*1* family and IS*903B* of the IS*5* family. Colistin combined with carbapenems (imipenem, meropenem), aminoglycosides (amikacin, gentamicin) or tigecycline was found to elicit in vitro synergistic effects against these colistin heteroresistant strains. Our experimental results showcase the potential of combination therapies for treatment of *K. pneumoniae* infections associated with colistin heteroresistance.

## 1. Introduction

*K. pneumoniae* is one of the most clinically significant Gram-negative bacteria that causes both community- and hospital-acquired infections, and it is associated with high mortality rates owing to its globally increasing resistance rates to multiple currently available antibiotics including carbapenems [1]. Currently, colistin treatment is considered as a salvage therapy for patients infected with multi-drug-resistant (MDR) Gram-negative bacteria such as *K. pneumoniae* [2]. MDR is defined as acquired non-susceptibility to at least one agent in three or more antimicrobial categories [3]. Consequently, due to its increasing usage in human medicine, reports of colistin resistance in *K. pneumoniae* have been on the rise [4,5,6,7,8,9]. Colistin, a cation polypeptide, was adopted as one of the last-line drugs for treating infections caused by MDR and XDR Gram-negative organisms. It works by binding to the negatively charged lipid A due to electrostatic interactions between the positively charged α,γ- diaminobutyric acid residues of colistin and the anionic phosphate groups of lipid A. Consequently, Mg^2+^ and Ca^2+^ are displaced from the phosphate groups of membrane lipids. This impairs the lipopolysaccharides, causing the outer membrane to become more permeable, which causes leakage of cytoplasmic content and eventually results in cell death [5,10]. Unfortunately, reports of colistin resistance mainly due to chromosomally encoded mechanisms have been on the rise. Resistance is mainly associated with mutations in the two-component regulatory systems, PmrAB and PhoPQ, or MgrB, which negatively regulates PhoPQ, and CrrAB, which regulates PmrAB. These mutations lead to the enzymatic substitutions of phosphate groups of lipid A with the cationic side chains, 4-amino-4-deoxy-L-arabinose and phosphoethanolamine, which lowers the net negative charge of the LPS and thus its affinity for binding to colistin [5,11]. Mobilizable insertion elements (ISs), particularly ISL*3* (IS*Kpn25*), IS*5* (IS*Kpn26*), IS*Kpn14*, and IS*903B*, were commonly identified to have been integrated into the *mgrB* gene in recent studies [12]. This mechanism of *mgrB* gene disruption by IS elements results in the development of colistin resistance in *K. pneumoniae* [13].

Several studies have highlighted that conventional MIC determination methods only detect homogenous resistant populations but fail to detect colistin heteroresistant strains (CH), which may lead to treatment failure [7]. Antibiotic heteroresistance is a phenotype in which a bacterial isolate contains resistant subpopulations of cells within an overall susceptible isolate which belongs to the same isogenic clone [14,15]. Heteroresistance to various classes of antimicrobials has been detected both in Gram-positive and in Gram-negative bacteria, and, most notably, against colistin in *K. pneumoniae* and in *Acinetobacter baumannii* [16,17]. During conventional antimicrobial susceptibility testing (AST), the presence of sporadic colonies growing within the inhibition area of a gradient strip or disc diffusion assay or the skipped wells observed in colistin broth microdilution assays of *Klebsiella* spp. and *Enterobacter* spp. can be used as a possible indicator of the presence of heteroresistance [15,18,19], although population analysis profiling (PAP) remains the reference method for the confirmation of heteroresistance [15]. Also, there is an increase in evidence suggesting the efficacy of drug combinations targeting colistin heteroresistance and multidrug heteroresistance for the treatment of infections caused by pandrug-resistant Gram-negative bacteria [20,21,22,23].

In our study, we used PAP to detect colistin heteroresistance, and we further analysed the underlying molecular mechanisms associated with heteroresistance among the identified CH *K. pneumoniae* strains. Also, we evaluated the potential of several drug combinations in preventing the regrowth of resistant subpopulations in CH strains via time–kill curve assays.

## 2. Materials and Methods

### 2.1. Bacterial Strain Collection and MIC Determination

We selected 55 colistin-susceptible (CS), unduplicated clinical isolates of MDR *K. pneumoniae* strains belonging to different sequence types from a collection of strains originating from specimens of intensive care unit patients in different countries in Europe as part of a multi-centre clinical trial from 1 December 2013 to 31 May 2017 (https://cordis.europa.eu/project/id/282512, accessed on 7 June 2023 NCT02208154; EU-FP7 RGNOSIS). MALDI-TOF (Bruker, Germany) was used to confirm the identification of the isolates to species level. Colistin MICs were determined by broth microdilution using Micronaut colistin-MIC strips (Merlin Diagnostika, Berlin, Germany) according to EUCAST 2023 (v 13.0) (Table 1). CS strains *E. coli* ATCC-25922, *P. aeruginosa* ATCC-27853, and colistin-resistant (CR) strains *E. coli* NCTC-13846 (*mcr-1* positive), *K. pneumoniae* 8400, were included as quality control strains for colistin MIC determination. Determination of MICs to other antimicrobial agents was performed via a quantitative gradient diffusion method (E-test, bioMérieux, Marcy-l’Étoile, France) (for gentamicin, imipenem, meropenem and tigecycline) and by broth microdilution (for amikacin) according to EUCAST 2023 (v 13.0) guidelines and clinical breakpoints (Table 1). *Escherichia coli* ATCC 25922 was used as quality control for amikacin, gentamicin, meropenem, imipenem and tigecycline MIC determination.

### 2.2. Detection of Colistin Heteroresistant Strains by Population Analysis Profiling (PAP) Assays

*K. pneumoniae* strains which displayed skipped wells in colistin broth microdilution assays as well as those found to be colistin-susceptible but with colistin MIC values close to the clinical resistance breakpoint (i.e., MIC of 2 mg/L) were considered as screening phenotypes indicating possible colistin heteroresistance, and were selected to undergo PAP assay. The strains were incubated overnight in Mueller Hinton Broth (MHB) (BD BBL^TM^, NJ, USA); then, 100 µL of 0.5 McF bacterial suspension was plated on a series of Mueller Hinton Agar (MHA) (BD BBL^TM^, NJ, USA) plates containing colistin in increasing concentrations (0.5 mg/L; 1 mg/L; 2 mg/L; 4 mg/L, 8 mg/L; 16 mg/L) [24]. The number of colonies were counted after 24 h of aerobic incubation of the plates at 37 °C, and a graph of the log_10_ CFU/mL was plotted against the increasing colistin concentrations. CH was defined as the detection of a resistant subpopulation which grew at a colistin concentration of ≥8× the MIC of the main susceptible population and at a frequency ≥1 × 10^−7^ [14]. *K. pneumoniae* ATCC 13883 (colistin-heteroresistant; MIC 1 mg/L) [25], *E. coli* ATCC-25922 (colistin-susceptible; MIC 0.25 mg/L), *K. pneumoniae* AN0222 (colistin-susceptible; MIC 0.25 mg/L) and *K. pneumoniae* 8400 (colistin-resistant; MIC 64 mg/L) were included as quality control strains in the PAP assays.

### 2.3. Time–Kill Assays

The time–kill kinetics of colistin alone or in association with several other drugs against the three confirmed CH *K. pneumoniae* strains (AN1505, IT0035, ITR0244) were assessed by adding colistin and other antibiotics to a broth bacterial culture in the logarithmic phase of growth containing approximately 10^6^ CFU/mL. The concentrations of all utilized antibiotics were adjusted to approach the peak serum concentration in patients receiving standard doses (colistin, 2 mg/L; tigecycline, 1 mg/L; amikacin, 64 mg/L; gentamicin, 16 mg/L; meropenem, 23 mg/L; imipenem, 26 mg/L) [20,26,27,28]. The cultures were incubated at 37 °C while being shaken at 220 rpm, and samples were collected after 0, 0.5, 2, 4, 8, 16 and 24 h of adding colistin. In total, 50 μL of the cultures and/or their serial dilutions was then spirally plated onto MHA plates, and the colonies were quantified after 24 h of aerobic incubation at 37 °C. Time–kill curves were then constructed by plotting the log_10_ CFU/mL against the various timepoints [23]. CH *K. pneumoniae* ATCC 13883 and CS *K. pneumoniae* AN0222 strain served as quality control strains.

### 2.4. Whole-Genome Sequencing and Analysis

Parental CH *K. pneumoniae* strains (AN1505, IT0035, ITR0244) and their resistant subpopulation colonies (AN1505-R1, AN1505-R2, IT0035-R1, IT0035-R2, ITR0244-R1, ITR0244-R2) were grown in MHB (Sigma Aldrich, Darmstadt, Germany). According to manufacturer instructions, genomic DNA was isolated using MasterPure Complete DNA and RNA Purification Kit (Epicentre, Madison, WI, USA). The isolated DNA was purified and concentrated with the Genomic DNA Clean & Concentrator-10 kit (Zymo, Irvine, CA, USA). Sequence libraries were then prepared using NexteraXT sample preparation kit, sequenced 2 × 250 bp, MiSeq (Illumina, San Diego, CA, USA). The quality of the raw sequences was assessed using FastQC, and primary analysis was performed using BacPipe v.1.2.6. CLC Genomics Workbench v. 9.5.1 (Qiagen, Hilden, Germany) was used to detect any previously validated modifications in the genes (*mgrB*, *phoP*, *phoQ*, *pmrB*, *pmrA*, *crrB*) which are known to confer colistin resistance. Insertion sequences were identified using ISfinder (https://isfinder.biotoul.fr/, accessed on 7 June 2023). All sequence data generated and analyzed in this study were deposited at NCBI under BioProject: PRJNA948355

## 3. Results

### 3.1. MIC Determination and Colistin Heteroresistance

A total of 16 colistin-susceptible *K. pneumoniae* strains out of the 55 initially selected isolates exhibited skipped wells during colistin broth microdilution MIC determination. Among these, three strains, namely ITR0244 (ST409), AN1505 (ST323) and IT0035 (ST15), were confirmed to be heteroresistant to colistin by PAP assay. These isolates, which were initially regarded as colistin susceptible together with *K. pneumoniae* ATCC 13883, a CH reference strain, had subpopulations growing in the presence of ≥2 mg/L of colistin. Resistant subpopulation colonies (AN1505-R1, AN1505-R2, ITR0244-R1, ITR0244-R2, IT0035-R1, IT0035-R2) were found to grow on plates with colistin concentration of 16 mg/L (Figure 1), whereas no resistant subpopulations were detected for the CS *E. coli* ATCC 25922 and CS *K. pneumoniae* AN0222. Both showed growth on plates with up to 1 mg/L of colistin. The MICs of the resistant subpopulations which grew on the plates were AN1505-R1, AN1505-R2: 32 mg/L; ITR0244-R1, ITR0244-R2: 64 mg/L; and IT0035-R1, IT0035-R2: 32 mg/L (Table 2).

### 3.2. Time–Kill Curve Assays

In the time–kill assay curves, colistin initially displayed a rapid bactericidal effect against all the strains, as evidenced by the (≥7 log_10_ to ≤3 log_10_ CFU/mL) decrease within 1 h of exposure (Figure 2). However, after 2–4 h of exposure to colistin, there was an increase in the number of colonies (≥7 log_10_) in all the CH strains except the CS strain AN0222 (Figure 2). This indicated that colistin was ineffective at completely eradicating these three CH strains due to the selection and enrichment of the resistant subpopulation after initially killing the susceptible population.

Next, the effect of colistin in combination with amikacin, gentamicin, meropenem, imipenem, or tigecyline was analysed using time–kill assay curves. In the colistin + gentamicin combination, the resistant subpopulation of CH strains was found to be eliminated for CH *K. pneumoniae* strains ITR0244 (Figure 3ii) and IT0035 (Figure 3iii), but not for the CH *K. pneumoniae* strain AN1505 for which regrowth of the colistin-resistant subpopulation occurred despite the presence of gentamicin (Figure 3i). In the colistin + amikacin, colistin + imipenem, colistin + meropenem and colistin + tigecycline combinations, the number of colonies decreased rapidly with time, and after 4 h there was no regrowth of resistant subpopulation colonies in any of the three strains, contrary to experiments in which they were exposed to colistin alone (Figure 4, Figure 5, Figure 6 and Figure 7). In the amikacin + colistin combination, a rapid decrease (≥7 log_10_ to ≤1 log_10_) and complete elimination of colonies were seen within 4 h in all the CH strains (Figure 4). However, when amikacin was used alone, the number of colonies decreased gradually with time except for the control CH strain (*K. pneumoniae* ATCC 13883). Similarly, when meropenem and imipenem was used along with colistin, a rapid decrease (≥7 log_10_ to ≤1 log_10_) of colonies can be seen, which was not the case when it was used alone (Figure 5 and Figure 6). Interestingly, CH strain ITR0244 was found to exhibit a heteroresistance growth pattern (Figure 5ii and Figure 6ii) when exposed to meropenem and imipenem alone, indicating a phenomenon of multi-drug heteroresistance. Similarly, CH strain AN1505 was also found to exhibit a heteroresistance growth pattern to meropenem in addition to colistin (Figure 6i). As expected, tigecyline when used alone did not affect the growth of the number of colonies at any point of time in any of the strains since the tigecycline MIC of the strains was greater or equivalent to the tigecyline antibiotic concentration (1 mg/L) used in the time–kill curve assay (Figure 7). However, when tigecycline was combined with colistin, it was found to provide synergistic effects, thus preventing resistant subpopulations in the CH strains.

### 3.3. Mutations Associated with Colistin Resistance Observed in the Heteroresistant Subpopulations

While all the three parental CH strains (ITR0244, AN1505, IT0035) were found to have an intact *mgrB* gene, their corresponding resistant subpopulation colonies (ITR0244-R1, ITR0244-R2, AN1505-R1, AN1505-R2, IT0035-R1, IT0035-R2) (ITR0244, IT0035, AN1505) were found to contain IS elements disruptions in the *mgrB* gene (Table 2). IS*903B* of IS*5* family was identified to be involved in the *mgrB* interruption of ITR0244-R1, AN1505-R1, AN1505R-2 and IT0035-R1. Also, IS elements IS*kpn34* of IS*3* family and IS*Kpn14* of IS*1* family was found to be involved in *mgrB* interruption of ITR0244-R2 and IT0035-R2, respectively.

## 4. Discussion

In our study, among the three MDR CH *K. pneumoniae* strains identified by PAP assays, two of them belong to ST15 and ST323, which are known multi-drug-resistant clones. Similarly, there are also other studies, where CH isolates belonging to well-known multi-drug-resistant international lineages such as ST11, ST307 were identified [24,29,30,31,32]. The association of colistin heteroresistance with isolates belonging to MDR clones which are known to be globally disseminated is very worrying since infections due to such strains are more likely to be treated with colistin if these are misclassified as susceptible by clinical diagnostic testing. The *mgrB* gene, which acts as the negative regulator of two component system PhoPQ was identified to be disrupted by IS elements IS*Kpn14* and IS*903B* in these CH-resistant subpopulatons. Hence, disruption of *mgrB* may result in overexpression of PhoPQ, which in turn is found to result in a subsequent increase in modifications of lipid A which confers colistin resistance [33]. Several studies have reported and validated the role of *mgrB* gene inactivation by point mutations, deletions or by insertion of IS elements in conferring colistin resistance [13,24,34].

The addition of colistin heteroresistance to the already significant public health threat of multi-drug resistance makes the development of new, combination treatment regimens more imperative. The in vitro time–kill curve assay results in this study demonstrated the potential of the CH strains to regrow significantly after 24 h after exposure to colistin. This could possibly be in line with the difficulty associated with treating infections caused by CH isolates, which may lead to clinical failure. The substantial regrowth after the initial rapid killing of the susceptible population also suggests that heteroresistance could progress to full-blown resistance with the continuous usage of colistin.

Here, we have evaluated the efficacy of colistin combined with different drugs to prevent the growth of resistant subpopulations with time. Using time–kill assay curves we show that the combination of colistin with aminoglycosides (amikacin, gentamicin), carbapenem (meropenem, imipenem) and tigecycline, a glycylcycline drug can effectively eliminate the growth of resistant subpopulations in such CH strains.

The high MIC values of the CH strains AN1505 and ITR0244 to gentamicin (MIC of 48 mg/L and of 24 mg/L, respectively) could be associated with the presence of *aac(3)-IIa* gene which is known to be associated with gentamicin resistance in *K. pneumoniae* [13]. Despite this, an in vitro synergistic effect of gentamicin with colistin resulted in the complete elimination of strain ITR0244. The presence of *aac(6′)-Ib-cr* in CH strains ITR0244, IT0035 and AN1505 resulted in the slow bactericidal activity of amikacin, thus resulting in gradual elimination of the subresistant population. However, the resistant subpopulations were found to be eliminated rapidly when amikacin was used in combination with colistin, indicating in vitro synergistic activity. The presence of a *bla*_OXA-48_ carbapenemase gene in the CH ITR0244 strain resulted in the emergence of carbapenem-resistant subpopulations when exposed to meropenem and imipenem alone. Thus, this CH ITR0244 strain proved to be carbapenem heteroresistant too. However, the resistant subpopulation growth of this strain exhibiting multi-drug heteroresistance was found to be prevented by the combination of colistin with any of the two carbapenem drugs. Hence, our study results are found to be in line with the recent reports suggesting the enhanced efficacy of antibiotic combinations to combat strains displaying multi-drug heteroresistance [23].

Similarly, several studies did also attempt to evaluate the potential of various drug associations with colistin in order to suppress colistin heteroresistance. In 2021, a study by Tian and Zhang reported that nearly half of the 95 carbapenem-resistant (CR) *K. pneumoniae* clinical isolates from a tertiary teaching hospital in China were heteroresistant to both tigeycline and polymyxin B. They also showed that the combination of tigecycline and polymyxin B was effective for preventing the resistant subpopulation in polymyxin B–tigecycline heteroresistant *K. pneumoniae* strains [21].

Another recent study reported a 72% prevalence of CH strains among 96 clinical CR *K. pneumoniae* isolates which were isolated from three tertiary hospitals in Beijing, China. Majority of the identified CH strains was found to belong to ST 11 (*n* = 69; 70.4%) followed by ST37 (*n* = 13; 12.3%). Additionally, colistin when combined with tetracyclines (minocycline and its glycylcycline derivative, tigecycline) or aminoglycosides (amikacin or gentamicin) was shown to produce in vitro synergistic effects in the identified carbapenem-resistant CH *K. pneumoniae* strains [22]. Though our study has also shown the prevention of the growth of the resistant subpopulation of CH strains by using different colistin combinations, the antibiotic concentrations used here were made to be in line with the peak serum concentration in patients receiving standard doses.

Recent guidelines from The Infectious Diseases Society of America (IDSA) recommends avoiding or restricting the usage of colistin and polymyxin B for the treatment of carbapenem-resistant Enterobacterales due to increased mortality and excess nephrotoxicity associated with polymyxin-based monotherapy [35]. However, the International Consensus Guidelines for the Optimal Use of the Polymyxins recommends combination therapy of polymyxin along with an additional antibiotic agent, preferentially with the one to which the infecting pathogen is susceptible [28]. Along similar lines, our results suggests that even in infections caused by CS strains, it is safer to prescribe colistin in any of these combinations while treating infections in order to eliminate the enrichment of the resistant subpopulation due to colistin exposure in case of CH strains. As the next steps, detailed in vivo studies are mandatory in order to optimize and implement these colistin combination therapies on a clinical scale.

## 5. Conclusions

In conclusion, our study highlights the potential benefit of using a combination of colistin with several other drug classes among which are carbapenems, aminoglycosides or tigecycline for the treatment of infections caused by CH *K. pneumoniae* strains. Pharmacokinetic and pharmacodynamic studies on these colistin combinations will be highly effective in devising strategies to combat CH treatment failures in hospitals. Also, the identified three clonally unrelated CH *K. pneumoniae* strains whose resistant subpopulations were found to harbor *mgrB* gene inactivation by IS elements, namely IS*Kpn14,* IS*1S* and IS*903B*. This phenomenon might indicate prior exposure of these strains to colistin pressure which might have induced the transposition event causing inactivation of *mgrB*. The results of the in vitro time–kill curve assays demonstrated the ineffectiveness of colistin alone to achieve complete elimination of CH strains with rapid regrowth of full-resistant populations following exposure; hence, there is a potentially high risk of treatment failure in vivo. Although our study suffers from a limitation that only a small number of CH strains was used, it highlights the need for further studies in order to understand how factors such as the proportion of resistant subpopulations and their degree of resistance can influence treatment outcomes. A more standardized definition and diagnostic techniques will aid in better detection of these strains and subsequently aid clinicians in selecting appropriate therapeutic options. Additional studies on the prevalence and worldwide distribution of CH strains are needed to further understand their epidemiological and clinical relevance.

## Figures and Tables

**Figure 1 genes-14-01426-f001:**
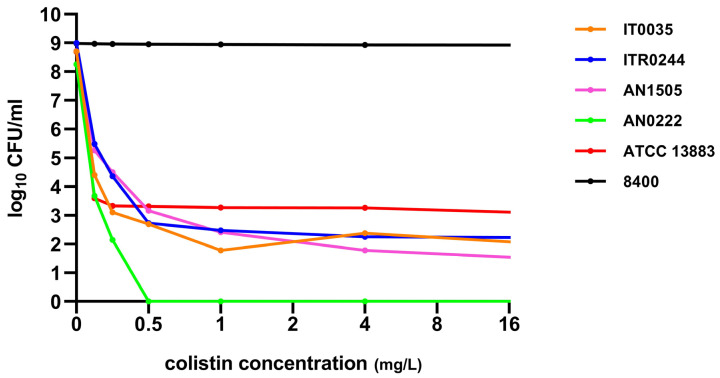
Population analysis profiles of the three CH strains (IT0035, ITR0244, AN1505), CS (AN0222), a CH reference strain (ATCC 13883), and a CR (8400).

**Figure 2 genes-14-01426-f002:**
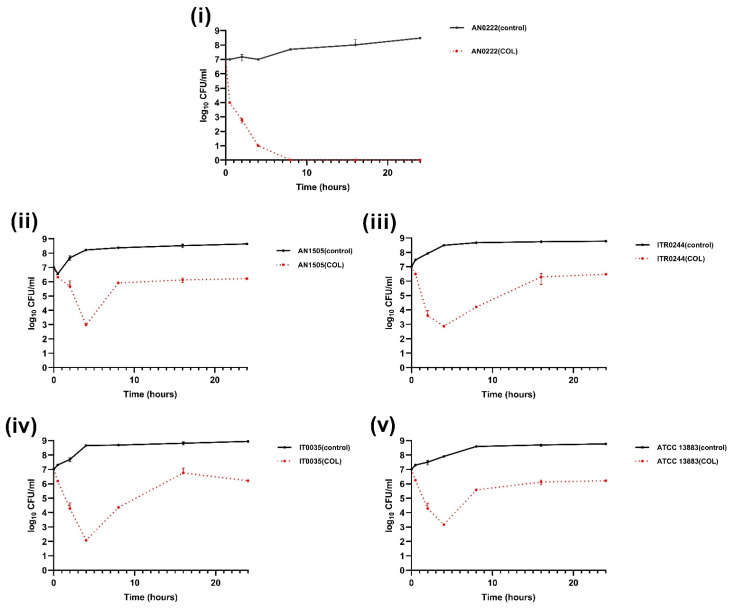
Time–kill curves of 2 mg/L colistin (COL). CS strain (**i**) AN0222 displayed sharp decrease in colony counts when exposed to colistin with no regrowth observed over time. CH strains (**ii**) AN1505, (**iii**) ITR0244, (**iv**) IT0035, (**v**) ATCC 13883 displayed an initial decrease followed by increasing colony counts (colistin resistant subpopulation) after 2–4 h of exposure to colistin.

**Figure 3 genes-14-01426-f003:**
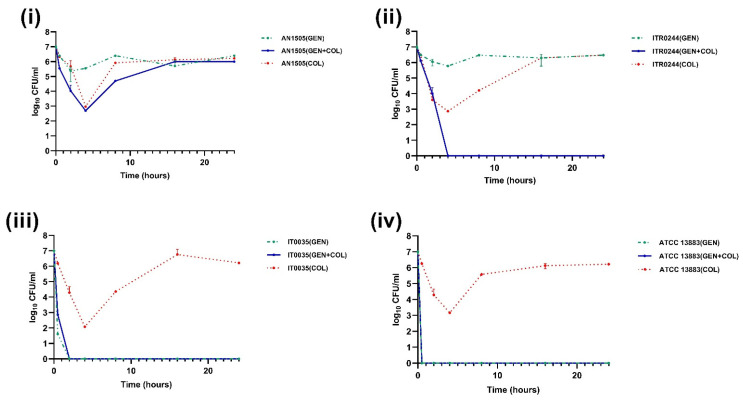
Time–kill curves of 2 mg/L colistin combined with 16 mg/L gentamicin (GEN). Except CH strain (**i**) AN1505, the resistant subpopulation was found to be eliminated in case of all other CH strains (**ii**) ITR0244, (**iii**) IT0035 and (**iv**) ATCC 13883 when exposed to colistin + gentamicin combination.

**Figure 4 genes-14-01426-f004:**
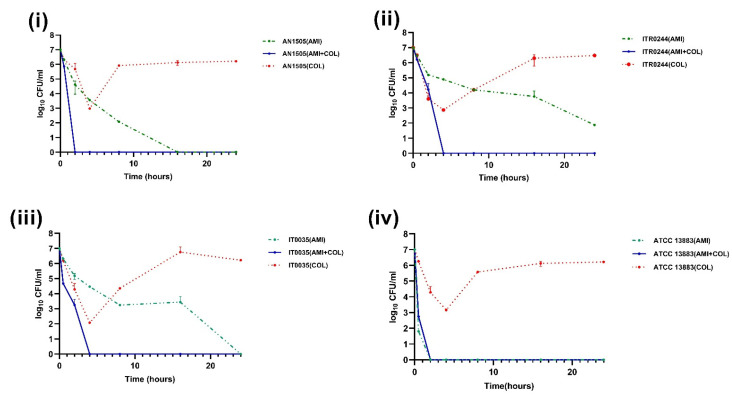
Time–kill curves of 2 mg/L colistin combined with 64 mg/L amikacin (AMI). The colistin resistant subpopulation was found to be eliminated in all the CH strains (**i**) AN1505, (**ii**) ITR0244, (**iii**) IT0035 and (**iv**) ATCC 13883 when exposed to colistin + amikacin combination.

**Figure 5 genes-14-01426-f005:**
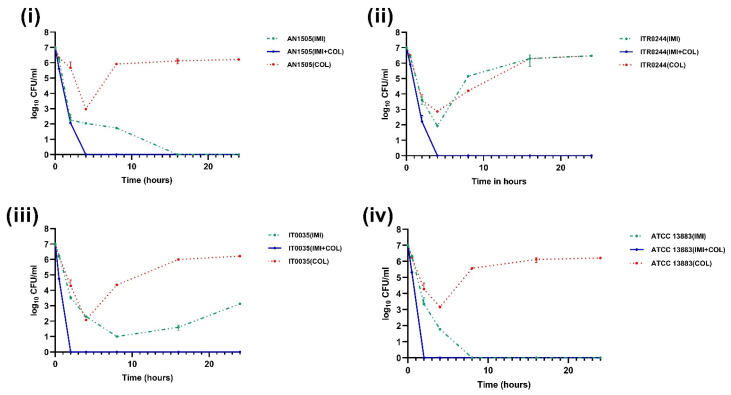
Time–kill curves of 2 mg/L colistin combined with 26 mg/L imipenem (IMI). The colistin resistant subpopulation was found to be eliminated in all the CH strains (**i**) AN1505, (**ii**) ITR0244 (**iii**) IT0035 and (**iv**) ATCC 13883 when exposed to colistin + imipenem combination.

**Figure 6 genes-14-01426-f006:**
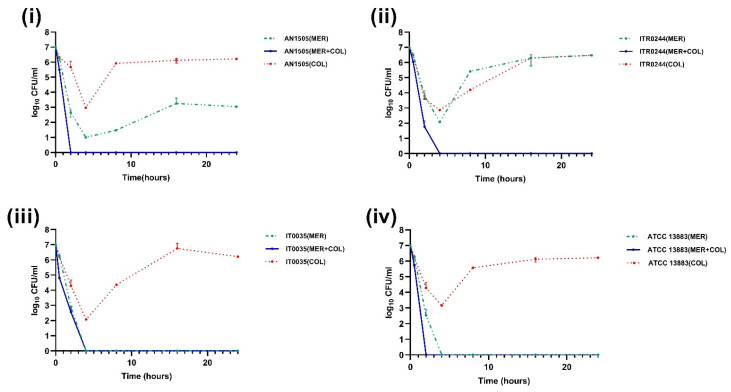
Time–kill curves of 2 mg/L colistin combined with 23 mg/L meropenem (MER). The colistin resistant subpopulation was found to be eliminated in case of all the CH strains (**i**) AN1505, (**ii**) ITR0244 (**iii**) IT0035 and (**iv**) ATCC 13883 when exposed to colistin + meropenem combination.

**Figure 7 genes-14-01426-f007:**
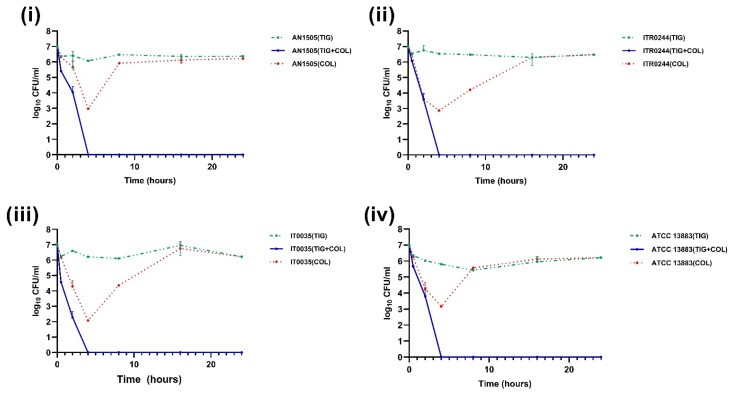
Time–kill curves of 2 mg/L colistin combined with 1 mg/L tigecycline (TIG). The colistin resistant subpopulation was found to be eliminated in all the CH strains (**i**) AN1505, (**ii**) ITR0244 (**iii**) IT0035 and (**iv**) ATCC 13883 when exposed to colistin + tigecycline combination.

**Table 1 genes-14-01426-t001:** MIC of CH strains against colistin (COL), gentamicin (GEN), amikacin (AMI), imipenem (IMI), meropenem (MER), tigecycline (TIG) and resistance genes (aminoglycoside and carbapenem) identified in them.

Strain	MLST	COL (mg/L)	GEN (mg/L)	AMI (mg/L)	IMI (mg/L)	MER (mg/L)	TIG (mg/L)	Resistome
		(S ≤ 2; R > 2)	(S ≤ 2; R > 2)	(S ≤ 8; R > 8)	(S ≤ 2; R > 4)	(S ≤ 2; R > 8)	ECOFF * 2 mg/L	
AN1505	ST323	1	48	4	0.25	0.05	1.5	*aac(3)-IIa*, *aac(6′)-Ib-cr*
ITR0244	ST409	1	24	2	0.75	0.75	4	*aac(3)-IIa*, *bla*_OXA-48_, *aac(6′)-Ib-cr*
IT0035	ST15	0.5	0.25	2	0.25	0.05	1.5	*aac(6′)-Ib-cr*
ATCC 13883	ST3	1	0.25	1	0.75	0.03	0.75	_

* No MIC breakpoint is currently defined for tigecycline. The epidemiological cut-off of tigecycline in susceptible wild-type *K. pneumoniae* isolates lacking resistance mechanisms is 2 mg/L.

**Table 2 genes-14-01426-t002:** MIC of resistant subpopulation of CH strains against colistin (COL) and colistin-associated genetic modifications identified in them.

Colistin-Resistant Subpopulation	Mutation	COL (mg/L)
AN1505-R1	*mgrB* gene interrupted by IS*903B* of IS*5* family at the 70th position	32
AN1505-R2	*mgrB* gene interrupted by IS*903B* of IS*5* family at the 70th position	32
ITR0244-R1	*mgrB* gene interrupted by IS*903B* of IS*5* family at the 117th position	64
ITR0244-R2	*mgrB* promoter region disrupted by IS*kpn34* of IS*3* family IS elements	64
IT0035-R1	*mgrB* gene interrupted by IS*903B* of IS*5* family at the 70th position	32
IT0035-R2	*mgrB* gene interrupted by IS*Kpn14* of IS*1* family at 123rd position	32

## Data Availability

All sequence data generated and analyzed in this study were deposited at NCBI under BioProject: PRJNA948355.

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
