# Peer review of "Insight into Antibiotic Synergy Combinations for Eliminating Colistin Heteroresistant Klebsiella pneumoniae"

_genes, 2023, doi:10.3390/genes14071426_

Round 1
Reviewer 1 Report
Excellent study. There is a need for continuing such studies with a larger number of isolates to confirm the findings and promote evidence based medicine.
Please define MDR Klebsiella pneumoniae. Then you could further state, that they appear to be ESBL/ AmpC but are tigecycline resistant with one isolate being resistant to genta also. Were the non CH isolates tigecycline resistant or sensitive? It is strange that tigecycline resistance is expressed by all the isolates. How do you explain that?
What about hetero-resistance to ANI 505 with Mem? I feel it also shows it.
Ehnglish is very good. Just few spelling mistakes.
Reviewer 2 Report
About the manuscript ID: genes-2444346 Title: Insight into antibiotic synergy combinations for eliminating colistin heteroresistant Klebsiella pneumoniae
Contains interesting and important results about the colistin heteroresitant Klebsiella pneumoniae isolates from clinical cases and the synergic effect with others antibiotics.
The manuscript is from my point of view, well written, however, need to improve few things.
1. Improve quality of graphs and use other colors to differentiate each assay, for example the green color of AN1505 and AN0222
2. Use italics for scientific names even on the legends of these graphs
3. Check spaces before or after a period or parenthesis
4. With previous and actual results of heteroresistance with colicin or other antibiotics, could be important to propose a hypothesis on how is the synergic between different actions at molecular level. Perhaps, the synergy between colicin and beta lactamases could be easier, however, for other antibiotics could be really interesting.
For me the English quality is good enough, because is easy to understand. Just few errors must be fixed.
Reviewer 3 Report
In their work, Rajakani et al. use state-of-the-art methods to identify colistin-heteroresistant K.pneumoniae isolates and the underlying genetic causes. In addition, they demonstrate that combination therapies may effectively eliminate heteroresistant strains. Their study is well designed and of interest for researchers and clinicians. I have few suggestions to improve the clarity of the manuscript.
L34: Please briefly introduce colistin resistance mechanisms.
L98: "in the logarithmic phase". How was the growth phase estimated?
L110: "their resistant subpopulation colonies (AN1505-R1, AN1505-R2, IT0035-R1, IT0035-R2, IT0244-R1, IT0244-110 R2) were grown in MHB". How were the resistant subcolonies R1 and R2 selected? Were these randomly chosen among the resistant colonies or did only two colonies per strain grow?
L119: "All the genomic modifications known to confer colistin resistance were analysed". Can you provide a list of the modifications that were investigated, or a reference describing those?
L134: "The MIC’s of the resistant subpopulations which grew on the plates were ..". Does this mean that the MIC testing was repeated for the R1 / R2 strains, revealing that these are no longer heteroresistant but show a defined MIC? Or do you mean that these subpopulations were initially recovered from plates with 32 or 64 mg/L colistin?
Table 2: “IS903B of IS5 family interruption at the 70th position of the mgrB gene” Please rephrase to clarify that mgrB was interrupted/disrupted (and not the IS element)
L189: Did the parental strains carry intact mgrB genes? This should be mentioned
L205: "identified to be disrupted by IS elements ISKpn14 and IS903B". Please introduce IS elements and their transposable character
L196: Can you give a hypothesis why only three strains acquired IS insertions in mgrB? Are the remaining strains free of IS elements that may insert into mgrB? Or is this a purely stochastic event, i.e., could heteroresistance occur in any Klebsiella strain that contains IS elements?
L234: Do you have a hypothesis for the underlying mechanism of the observed carbapenem heteroresistance?
The manuscript is well written and structured.
